# Synovial Fluid and Serum MicroRNA Signatures in Equine Osteoarthritis

**DOI:** 10.3390/ijms262211190

**Published:** 2025-11-19

**Authors:** Catarina I. G. D. Castanheira, Sarah Taylor, Eva Skiöldebrand, Luis M. Rubio-Martinez, Matthias Hackl, Peter D. Clegg, Mandy J. Peffers

**Affiliations:** 1Department of Musculoskeletal and Ageing Research, Institute of Life Course and Medical Sciences, University of Liverpool, Liverpool L7 8TX, UK; 2The Royal (Dick) School of Veterinary Studies, The Roslin Institute, University of Edinburgh, Edinburgh EH8 9YL, UK; 3Swedish University of Agricultural Sciences, SE-750 07 Uppsala, Sweden; 4Sussex Equine Hospital, Billingshurst Road, Ashington RH20 3BB, UK; 5TAmiRNA GmbH, Ltd., 1110 Vienna, Austria

**Keywords:** biomarkers, equine, microRNA, osteoarthritis, serum, small RNA sequencing, synovial fluid

## Abstract

The aim of this study was to identify differentially expressed microRNAs (miRNAs) in serum and synovial fluid (SF) samples of control horses and those with osteoarthritis (OA) to identify potential candidates for biomarkers of disease. Total RNA was extracted from serum and SF samples of control (n = 4) and OA (n = 9) horses and sequenced. Differential expression analysis, pathway analysis and miRNA target prediction were performed. A group of six miRNAs (eca-miR-199a-3p, eca-miR-148a, eca-miR-99b, eca-miR-146a, eca-miR-423-5p and eca-miR-23b) was selected for validation in an independent cohort (serum, n = 46; SF, n = 88). The effect of clinical variables on miRNA expression was also assessed. Sequencing analyses found 43 and 23 differentially expressed miRNAs in serum and SF samples, respectively. Pathway analysis showed miRNAs were involved in inflammatory disease/response and associated with OA pathways. miRNA expression in serum was strongly associated with the horses’ workload, while age had a pronounced influence on miRNA expression in SF. Distinct patterns of miRNA differential expression were observed in serum and SF samples from horses with OA compared to controls. miR-199a-3p and miR-148a warrant further investigation as potential biomarkers of equine OA. Further characterization of these molecular changes could provide novel insights into the mechanisms of early OA.

## 1. Introduction

Genetic research has long put RNA at the center of the field of molecular biology, yet the importance of the regulatory functions of non-coding RNAs for organismal development has only been brought to light in the past two decades [1]. The transcriptional landscape is comprised of several RNA classes with varying functions and is broadly separated into coding RNAs, long non-coding RNAs (lncRNAs) and small non-coding RNAs (sncRNAs). The sncRNA group includes transfer RNAs (tRNAs), which are involved in protein translation; small nuclear RNAs (snRNAs), which are involved in splicing events; small nucleolar RNAs (snoRNAs), which are mainly involved in the modification of other RNAs; and short regulatory non-coding RNAs, such as piwi-associated RNAs (piRNAs), small interfering RNAs (siRNAs) and microRNAs (miRNAs), which regulate gene expression [2]. Among these, the miRNA class is the best characterized. These evolutionarily conserved molecules are, on average, 22 nucleotides in length and participate in the post-transcriptional regulation of genes [3]. miRNAs target messenger RNAs (mRNAs) and generally repress their translation or promote mRNA decay [3]. On rare occasions, miRNAs can also increase the expression of specific mRNAs [4]. miRNAs are important contributors to many biological functions, including cell differentiation, embryogenesis, metabolism and organogenesis. Of particular relevance is their contribution to intercellular communication [3]: miRNAs influence multiple physiological and pathophysiological processes, and their expression can be altered as a result of cellular damage and tissue injury [5]. Therefore, miRNA “signatures” can be detected both in damaged tissues and in the circulation in biofluids, including blood or synovial fluid (SF) [5]. This, along with the fact that they are remarkably stable in the circulation, makes miRNAs prime candidates for use as non-invasive biomarkers in molecular diagnostics of disease [5].

The inability to detect pre-clinical changes in osteoarthritis (OA) is one of the main barriers to the development of effective therapies against this disease. OA is one of the most common causes of lameness in horses and a significant welfare concern due to pain and disability [6,7]. Currently, when a clinical diagnosis of OA is reached, the changes that have occurred in the joints are generally advanced and irreversible, often leading to early retirement, and can lead to euthanasia of these animals. Therefore, developing a technique that allows for early diagnosis and intervention is a major goal of OA research [8]. miRNAs are promising candidates in the search for diagnostic biomarkers of OA and have been extensively investigated in human studies [8]. Previous work has demonstrated that miRNAs are involved in the regulation of cell apoptosis, inflammation and chondrocyte homeostasis and metabolism, displaying protective or destructive roles and, sometimes, both [9]. Furthermore, researchers have investigated the levels of circulating miRNAs in OA patients and found differentially expressed molecules with potential clinical use as biomarkers of disease and as predictors of OA severity [10,11]. For example, miR-140-3p, miR-33b-3p and miR-671-3p, which are downregulated in the articular cartilage of human OA patients compared with healthy patients, were also found to be downregulated in the serum of these OA patients, showing potential as diagnostic tools [12]. Another example is miR-378a-5p, which is commonly detectable in SF in late-stage OA and mostly undetectable early-stage OA, showing a potential for disease stratification [13]. Additionally, our group previously found a pattern of differentially expressed sncRNAs in equine OA SF linked to OA pathogenesis, which included miR-23b, let-7a-2 and miR-223 [14].

The objective of the present study was to determine the miRNA profile of serum and SF of horses affected by OA and control horses, identify differentially expressed miRNAs between the two groups and determine their potential as biomarkers of OA.

## 2. Results

This study used a small RNA sequencing approach to investigate the miRNA profile of matched (i.e., from the same horse) equine serum and SF in OA (Figure 1). Briefly, in stage 1 (exploration stage), matched samples of serum and SF were sequenced and analyzed, and a panel of six miRNAs was selected for validation. In stage 2 (validation stage), gene expression analysis of the selected group of miRNAs was carried out in the sequencing cohort and in an independent cohort (validation cohort).

### 2.1. Stage 1: Exploration Stage

#### 2.1.1. Sample and Group Characterization in the Sequencing Cohort

A total of 13 matched samples of serum and SF (OA, n = 9; control, n = 4) were collected for sequencing (Table 1). The horses’ age and proportion of males (neutered and non-neutered)/females were similar among groups (*p* = 0.605 and *p* = 0.500, respectively); however, the absolute number of males included in the entire sequencing cohort was double that of females (n = 8 and n = 4, respectively). Most SF samples were collected from carpal joints. For most horses with available data in the OA group (60%), OA was classified as mild; the details on sample classification are described in the methods.

#### 2.1.2. Sequencing Data Overview

The average number of reads was similar for both serum and SF (mean 48 million and 47 million reads/sample, respectively). Several types of RNA molecules were identified, including both small and long RNA types (Figure 2). The relative abundance of miRNAs varied between 0.04 and 57.03% in serum and 0.03 and 33.64% in SF (Appendix A).

A total of 308 miRNAs were identified in all samples, with 298 identified in serum and 203 in SF (Appendix A). Overall, 105 miRNAs were unique to serum, and 10 were uniquely identified in SF, with most of these molecules found in OA samples.

A heatmap was built on reads per million (RPM) normalized and scaled reads using the unit variance method for visualization in heatmaps (Figure 3A). This unsupervised analysis demonstrated a generally distinct miRNA expression pattern between serum and SF samples, with all but one sample (95SF) clustering according to sample type. Of note, 95SF was the sample with the highest percentage of unclassified genome (Figure 2), which may have affected sample clustering.

Principal component analyses (PCA) were performed using RPM normalized miRNA reads, which were scaled using unit variance. Principal components (PCs) were calculated using singular value decomposition using the function prcomp() in R (package stats v3.6.2) and visualized using the package ggplot2 v.3.4.2 (Figure 3B). All samples were retained to reflect the full data structure, and no specific outlier removal processes were applied. The PCA plots showed an overlap between the OA and control groups for both serum and SF.

#### 2.1.3. Differential Expression Analysis

In serum, 43 miRNAs were differentially expressed (*p* < 0.05) between the OA and control groups (Table 2); of these, 15 had a false discovery rate (FDR)-adjusted *p* < 0.05. In SF, there were 23 differentially expressed miRNAs between the OA and control groups, of which seven had an FDR-adjusted *p* < 0.05. Nine differentially expressed miRNAs were common for serum and SF: eca-miR-1291a, eca-miR-148a, eca-miR-1892, eca-miR-199a-3p, eca-miR-199b-3p, eca-miR-206, eca-miR-23b, eca-miR-27b and eca-miR-423-5p. For most of these molecules, the directionality of change was similar in serum and SF.

#### 2.1.4. Target Prediction and Pathway Analysis

Target prediction revealed 104 experimentally validated targets for 22 of the differentially expressed miRNAs. Ingenuity Pathway Analysis (IPA; Qiagen, Manchester, UK) Core Analysis of the combined list (interactome) of differentially expressed miRNAs and miRNA targets revealed involvement of inflammatory disease and response (Appendix A). Some of the diseases and functions predicted to be associated with the differentially expressed miRNAs included osteoarthritis (*p* = 2.93 × 10^−3^), rheumatic disease (*p* = 8.11 × 10^−5^) and atrophy of skeletal muscle (*p* = 9.45 × 10^−3^; Appendix A).

#### 2.1.5. Candidate Biomarker miRNAs

Six miRNAs were selected for further validation using the criteria described in the Methods. These were eca-miR-199a-3p, eca-miR-148a, eca-miR-99b and eca-miR-146a, which were increased in the OA vs. control group; and eca-miR-423-5p and eca-miR-23b, which were decreased in the OA vs. control group (Table 2; Appendix A).

### 2.2. Stage 2: Validation Stage

#### 2.2.1. Sample and Group Characterization in the Validation Cohort

A total of 46 serum samples (control, n = 23; OA, n = 23) and 88 SF samples (control, n = 44; OA, n = 44) were collected for the validation cohort (Table 3).

For serum samples, there were no significant differences in mean age or sex between the control and OA groups (*p* = 0.6582 and *p* = 0.2603, respectively). There were more racing horses in the OA group than horses of any other occupation (9/18 with available information); in contrast, there were no racing horses in the control group. The horses’ level of work was significantly different among groups (*p* = 0.0214), with 56.6% vs. 8.7% of horses being described as having a work level of 3 (intense work) in the OA vs. control group, respectively.

For SF samples, mean age (SD) was significantly higher in the OA vs. control group (*p* = 0.0003). The OA group had a significantly higher proportion of neutered males vs. females than the control group (69.2% vs. 32.0%, respectively). There were more Thoroughbred horses in each group than horses of any other breed. Mean joint macroscopic score and mean articular cartilage microscopic score were higher in the OA vs. control group.

#### 2.2.2. miRNA Expression Analysis

Relative expression of the previously selected, candidate biomarker miRNAs were analyzed using RT-qPCR in the sequencing and the validation cohorts.

In serum, miRNA expression in the sequencing cohort (Figure 4A) followed the same trend as the sequencing data (Appendix A) for all miRNAs except eca-miR-23b. However, none of the differences between the control and OA groups were statistically significant. In the validation cohort (Figure 4B), expression of eca-miR-199a-3p, eca-miR-99b and eca-miR-23b was numerically higher in the OA vs. control group, which was in line with sequencing data (Appendix A). In contrast, expression of eca-miR-148a and eca-miR-423-5p was generally similar among groups, while eca-miR-146a was numerically decreased in the OA group, which differed from the sequencing data. None of these differences reached statistical significance.

In SF samples, miRNA expression in the sequencing cohort (Figure 4C) was generally very low in the control group, with one miRNA (eca-miR-99b) not being detected in any of the samples. No statistically significant differences were found between the control and OA groups. In the SF validation cohort (Figure 4D), eca-miR-199a-3p expression was numerically increased in the OA vs. control group, which was in line with the sequencing data; however, these differences were not statistically significant. For the remaining miRNAs, expression was generally similar among groups.

#### 2.2.3. Influence of Clinical Variables in miRNA Expression

In serum, miRNA expression was significantly associated with level of work (FDR *p*-value = 0.01 [PC1]) and group category (FDR *p*-value = 0.02 [PC6]; Appendix A). In SF, miRNA expression was significantly associated with age (FDR *p*-value = 0.01 [PC1]), group category (FDR *p*-value = 0.01 [PC4]) and joint macroscopic score (FDR *p*-value = 0.01 [PC4]; Appendix A).

The associations between miRNA expression and level of work and age were further explored (Appendix A). In serum, samples from horses with a lower level of work (0–2) appeared to cluster separately from those with higher levels of work (3). In SF, there were two distinct clusters separated by PC1; while this separation could not be solely explained by age, samples from younger horses were more abundant in the left cluster (Appendix A). To further investigate the influence of age on miRNA expression in SF, horses were separated into four age classes: <6 years (n = 27), 6–12 years (n = 8), 13–20 years (n = 37) and >20 years (n = 9); horses of unknown age were excluded from further analyses. Results showed that most SF samples in the <6 age class (19/27) were in the control group, and most samples in the 13–20 age class (23/37) were in the OA group. For the selected panel of miRNAs, expression was generally numerically highest for horses <6 years old (Appendix A).

## 3. Discussion

With advances in molecular biomarker research offering the potential for new diagnostic tools, OA research has been shifting its focus from disease management to detection and prevention of early disease [17]. This study investigated the miRNA profile of equine serum and SF in OA and control groups and identified a panel of 43 miRNAs that were differentially expressed in serum as well as 23 miRNAs that were differentially expressed in SF. Pathway analysis revealed that altered miRNAs were possibly involved in inflammatory responses, cell apoptosis, maturation of chondrocytes and atrophy of skeletal muscle, with a significant association with OA. Additionally, the level of work and age were identified as the main clinical variables influencing miRNA expression in serum and SF, respectively.

Circulating miRNAs in blood are attractive candidates as biomarkers of disease because samples can generally be obtained through minimally invasive techniques and provide an overview of systemic events. On the other hand, SF is a plasma ultrafiltrate that closely reflects the local processes in the joint due to its biological and functional proximity with the different joint tissues [18]. This, along with the fact that SF can be sampled with relative ease from most equine synovial joints, renders SF analysis a valuable diagnostic tool for OA in clinical settings [16]. To compare the molecular patterns between SF and serum, matched samples were collected for each horse in the sequencing cohort. The results showed a wide variety of small RNA molecules in serum and SF samples, which is in line with previous findings [14,19]. The relative abundance of miRNAs was highly variable but generally higher in serum samples compared with SF samples. This difference may be due to the systemic origin of the miRNAs present in the serum, as opposed to the more local, likely articular origin of the miRNAs contained in SF.

Unsupervised analysis of miRNA revealed some overlap among OA and control samples. Still, there were 111 miRNAs exclusively present in OA samples, of which 73 were exclusive to serum, 10 were exclusive to SF, and 28 were present in both serum and SF, highlighting the differences in miRNA expression among groups. Sequencing data analysis revealed a total of 43 differentially expressed miRNAs in serum and 23 in SF between the OA and control groups. Of these, all but 18 miRNAs (miR-1291a, miR-1892, miR-2483, miR-30e, miR-342-5p, miR-423-3p, miR-490-3p, miR-598, miR-615-5p, miR-7177b, miR-744, miR-8951, miR-8954, miR-8977, miR-8986b, miR-8992, miR-9048, miR-92b) had been previously linked to OA in some capacity [14,20,21,22,23,24,25,26,27].

Bioinformatic analysis of the differentially expressed miRNAs revealed a significant association with inflammatory disease and response, while target prediction revealed gene associations to osteochondral biology. Together, these results suggest that the differentially expressed miRNAs in OA joints may be responsive to or drivers of inflammatory events and/or cartilage degradation, repair and subchondral bone remodeling.

In this study, miR-199a-3p expression was numerically increased in the OA vs. control group (not statistically significant) in both cohorts and sample types. This is in line with the results from an experimental model of equine OA that reported miR-199b-3p (which has the same mature sequence as eca-miR-199a-3p) to be increased in the SF of OA vs. control joints [28]. miR-199a-3p appears to be implicated in chondrogenic processes, being upregulated during chondrogenesis of human adipose-derived stem cells [27] and downregulated in interleukin (IL)-1β-treated chondrocytes [29]. Additionally, chondrocyte transfection with miR-199a-3p mimics decreased expression of collagen type (COL2), aggrecan and SOX9 [30]. In this study, miR-199a-3p was upregulated in young (<6 years) horses when compared with middle-aged (13–20 years) horses; additionally, within the 13–20 years age class, miR-199a-3p expression was upregulated in OA vs. control samples. These findings indicate that, while miR-199a-3p is a promising candidate as a biomarker of OA, age is a cofounding factor for its expression and analysis.

Expression of eca-miR-148a was increased in OA serum and SF when compared with the control, and RT-qPCR analysis confirmed these results (albeit not statistically significant and despite differences not being as marked in the validation cohorts). eca-miR-148a is homologous to hsa-miR-148a-3p, which is thought to have a role in cartilage regeneration and is decreased in OA cartilage compared with healthy cartilage [31]. Overexpression of miR-148a in human chondrocytes resulted in increased extracellular matrix deposition and promoted a downregulation of metalloproteinase (MMP) 13 and A disintegrin and metalloproteinase with thrombospondin type 1 motif 5 (ADAMTS5) [31]. Due to these chondroprotective functions, it is possible that the increased expression of miR-148a in SF and serum detected in the present study occurred in response to injury in an attempt to prevent further tissue alterations. Interestingly, age also appeared to affect miR-148a expression, which was particularly evident for the horses in the SF validation cohort between 13 and 20 years old.

Previous studies found miR-99b-5p to be upregulated in OA and demonstrated that overexpression of this miRNA in chondrocytes increased MMP13 and senescence-related factors and decreased COL2 [32]. While the precise role of miR-99b-5p is not yet known, this molecule is part of an evolutionarily conserved cluster (miR-99b/let-7e/miR-125a) that is highly upregulated during the early stages of osteoclastogenesis and has a direct relationship with Nuclear Factor kappa B [33]. However, eca-miR-99b expression was consistently low in the validation stage of this study, both in serum and SF; these results suggest that eca-miR-99b might not be an appropriate biomarker for equine OA. In this study, eca-miR-146a-5p was increased in serum in the OA group when compared with the control in the sequencing cohort; however, these results were not consistent in the validation cohorts. In previous reports, miR-146a played a protective role in OA [34], evidenced by a reduction in reduced IL-1β and induced tumor necrosis factor α production in human chondrocytes overexpressing miR-146-5p [35]. Remarkably, in the present study, expression of miR-146a-5p was statistically upregulated in the SF of young (<6 years) horses when compared with all other age classes, revealing a strong association between age and miRNA expression. Furthermore, analysis within the 13–20 years age class showed a distinction in miRNA expression between OA and control samples.

miR-423-5p has been previously identified as a negative regulator of osteoblastogenesis [36,37], and women with bone pathologies display higher serum expression of miR-423-5p than controls [23]. However, a study looking at miRNA expression in cartilage of OA patients used miR-423-5p, a reference gene due to its stability across both OA and control groups [38]. While this miRNA might be implicated in OA pathogenesis, further research is needed to elucidate its precise roles. In the validation stage of the present study, miR-423-5p expression in serum was increased in the control vs. OA group (not statistically significant) but was similar between groups in SF. miR-423–5p was one of the few miRNAs whose expression was not increased in younger (<6 years) horses compared with older horses. Finally, analysis of sequencing data revealed eca-miR-23b expression as decreased in OA compared with the control. This was an unexpected finding, as previous studies have found this miRNA to be upregulated in OA tissues [39,40]. However, validation results for eca-miR-23b were inconsistent with sequencing results, and this miRNA was generally upregulated in OA compared with the control in both cohorts and both sample types. While sequencing is a powerful tool, it is subject to bias, particularly during library amplification [41]; it is possible that this was the case for eca-miR-23b.

Clinical characteristics and lifestyle factors are known to influence different epigenetic mechanisms, including miRNA expression [42]. This study investigated the influence of different clinical variables in the expression of a selected group of miRNAs and found that the level of exercise placed upon horses is significantly associated with variations in miRNA expression in their serum. In accordance with these findings, previous studies have shown that exercise is associated with alterations in circulating miRNAs in horses [43,44,45]. A study investigating the miRNA population of plasma extracellular particles in horses reported increased levels of eca-miR-486-5p during and after an endurance race and decreasing levels of eca-miR-9083 [45]. Interestingly, eca-miR-486-5p was identified in the present study as one of the most stable miRNAs in serum and SF; further studies may help elucidate the precise role of this miRNA. Age was also found to be significantly associated with miRNA levels in SF in this study. Numerous studies have previously described variations in miRNA patterns with age [46], including in joint tissues [47,48]. This is particularly relevant for OA studies because age is a critical factor in OA development [49]. In the present study, most molecules were upregulated in younger horses (<6 years) when compared with older horses. These findings highlight the importance of accounting for variations in age in OA studies, especially if attempting to validate potential OA biomarkers.

Achieving robustness and reproducibility of molecular biomarkers across different populations, settings and laboratories is a major challenge. miRNA profiling data is only partially reproducible between different platforms, and even within the same platform, variation is common [50]. In this study, it was not possible to validate the sequencing findings at an FDR-adjusted *p* < 0.05, which might have been related with the distinct set of samples used in sequencing and in validation. Horses in the sequencing cohort were generally younger than those in the validation cohort. In young horses, OA may arise as a consequence of developmental orthopedic diseases, such as osteochondrosis and osteochondritis dissecans. Meanwhile, in older horses, OA may develop gradually with age through degeneration and destruction of cartilage and other joint tissues. Additionally, for animals from a racing background, OA alterations can arise after repetitive microtrauma. We hypothesize that differences in etiopathogenesis of OA can affect the miRNA patterns in the joints. Furthermore, given that the level and type of exercise may be associated with alterations in miRNA expression in serum, it is possible that the type of work and the associated injuries indirectly influence miRNA regulatory mechanisms and can partially explain the disparity in the results of this study.

There were also limitations associated with the sample collection and group allocation. For abattoir-derived samples, macroscopic and microscopic joint assessments were performed, yet clinical data such as lameness could not be evaluated. The relationship between joint macroscopic changes and overt clinical signs remains uncertain, and in abattoir specimens where clinical history is lacking, we cannot reliably assess how those macroscopic changes align with clinical lameness. Additionally, there is a potential for degradation and/or altered biochemical profiles in abattoir-derived samples, which can also affect the results. On the other hand, it is not possible to collect information regarding macroscopic and microscopic joint assessments for excess clinical samples. All of these issues can lead to inconsistencies in how the samples are allocated to the different groups and potentially lead to different subpopulations within the control and OA groups.

The SF samples in this study were collected from different joints, and it is currently unclear whether different joints inherently exhibit different miRNA profiles, which could increase miRNA variability and affect the validation results. Furthermore, there was no information available on prior treatment/medication, and its potential effect on miRNA expression could not be assessed. An additional limitation of this study was the small set of molecules selected for validation, which was restricted by finances. Restricting the selection to miRNAs that were differentially expressed in both serum and SF may have limited the findings, as some molecules present exclusively in one sample type could still possess valuable biomarker properties. Future studies investigating the miRNA and the overall sncRNA profile of equine osteoarthritis would benefit from analyzing larger cohorts of patients with similar clinical characteristics, particularly age and level of work. Furthermore, investigating information from a panel of multiple biomarkers as opposed to a single biomarker is more likely to achieve better performance [51].

## 4. Methods

### 4.1. Sample Collection

#### 4.1.1. Sequencing Cohort

Post-mortem samples (matched serum + SF) were collected from racehorses slaughtered at an abattoir in Sweden and processed by a veterinary surgeon as previously described [15]. Upon collection, samples were centrifuged at 1800 *g* for 20 min to remove cellular debris, snap frozen in liquid nitrogen and stored at −70 °C until further processing.

Clinical samples (matched serum + SF) consisted of excess samples obtained by veterinary surgeons during lameness examinations in an equine hospital in the UK. Serum samples were obtained through centrifugation of blood samples at 2000 *g* for 20 min at 4 °C after clotting and stored at −80 °C. SF was sterilely obtained from the joints being assessed by the clinician and was processed as previously described [14].

#### 4.1.2. Validation Cohort

Serum samples consisted of excess clinical samples obtained by veterinary surgeons in one equine clinic in the UK prior to routine dental examination and two other centers (UK and China) either during lameness examination or at post-mortem. Samples were collected and processed similarly to the sequencing cohort.

SF was collected at post-mortem from the metacarpophalangeal joints of horses from an abattoir in the UK as a by-product of the agricultural industry or from horses donated to research in an equine hospital in the UK. Briefly, the joints were aseptically dissected to collect SF, and samples were processed similarly to the sequencing cohort.

#### 4.1.3. Ethical Considerations

Samples from the Swedish abattoir were collected in accordance with local legislation. For samples collected from the UK abattoir, ethical review and approval were waived due to samples being obtained as a by-product of the agricultural industry—the Animal (Scientific procedures) Act 1986, Schedule 2, does not define collection from these sources as scientific procedures, and ethical approval was, therefore, not required.

Ethical collection of excess clinical samples was approved by the University of Liverpool’s Veterinary Research Ethics Committee (VREC660a (2020.08.19)). Hong Kong Jockey Club samples were collected under the regulations of the club with owner consent.

### 4.2. Group Allocation

Samples were assigned to each group (control and OA) based on the absence or presence of a diagnosis and/or signs of OA. The specific criteria for group allocation varied depending on the type of samples collected, as described in the following sections.

#### 4.2.1. Sequencing Cohort

For post-mortem collections (serum and SF), the left middle carpal joint of the corresponding animal was dissected, and the articular cartilage of the proximal surface of the carpal bones in the proximal row was inspected macroscopically [15]. Horses with no evidence of macroscopic lesions were included in the control group. Horses with macroscopic lesions were included in the OA group. The area of existing lesions was measured using a digitized image system (Kontron System 100, Kontron, Ismaning, Germany), and the severity of disease was classified according to the size of lesions: if a cartilage area smaller than 40 mm^2^ was affected, OA was classified as mild; if lesions extended over an area between 40 mm^2^ and 230 mm^2^, OA was considered moderate; and for horses exhibiting lesions on an area exciding 400 mm^2^, OA was considered severe [15].

For excess clinical samples (serum and SF), groups were based on the clinical diagnosis made by the attending veterinary surgeon. Diagnosis of OA was made through a combination of medical history, presence of clinical signs such as lameness, joint effusion or pain on joint flexion, positive response to intra-articular anesthesia, presence of osteoarthritic changes on imaging exams such as radiography and the exclusion of differential diagnoses. Horses diagnosed with OA were included in the OA group; horses that were not diagnosed with OA and did not present any signs of joint infection were included in the control group. The severity of OA was not classified.

#### 4.2.2. Validation Cohort

Group allocation of excess clinical samples varied depending on sample and collection type. Serum samples were all excess clinical samples and were allocated into groups based on owner reported clinical history: if owners reported their horse to have a diagnosis of OA made by a veterinary surgeon, samples were included in the OA group; if owners reported no history of OA, samples were included in the control group; cases in which owners were unsure about an OA diagnosis but reported a history of joint corticosteroid injections or chronic intermittent lameness of unknown origin were also included in the OA group.

For SF collected as excess clinical samples, group allocation was based on the clinical diagnosis made by the attending veterinary surgeon, similarly to the sequencing cohort. For SF collected at post-mortem, joints were dissected and scored by two independent researchers using the Osteoarthritis Research Society International (OARSI) equine macroscopic grading system [52]. Samples were assigned to groups based on the average macroscopic score, where horses with scores of 0–1 were included in the control group and >3 were included in the OA group.

### 4.3. Demographics

Information regarding age, sex, breed, collection site (abattoir vs. hospital/clinical), BCS, occupation, level of work, presence of shoes and joint histological/macroscopic scoring was collected. Information regarding disease severity and OA-affected joints was also collected for animals in the OA group. Demographic data were analyzed descriptively and compared between groups; statistical tests were selected after normality testing using GraphPad Prism version 8.0.

### 4.4. Small RNA Sequencing

#### 4.4.1. RNA Extraction

SF samples were pre-treated with hyaluronidase (H3884, Sigma-Aldrich, Gillingham, UK) and filtered through a Costar^®^ Spin-X^®^ polypropylene microcentrifuge tube filter with a 0.22 μm pore cellulose acetate membrane (Corning, Flintshire, UK). Serum was pre-processed through centrifugation to remove cryoprecipitates.

Total RNA was extracted from a 200 μL sample using the miRNeasy Serum/Plasma Advanced Kit (Qiagen, Manchester, UK) with the addition of glycogen (ThermoFisher Scientific, Paisley, UK), and the RNA was eluted in 18 μL of RNase-free water.

#### 4.4.2. Library Preparation and Sequencing

Small RNA library preparation was undertaken with the CleanTag^®^ Small RNA Library Preparation kit (TriLink Biotechnologies, San Diego, CA, USA), according to manufacturer’s instructions, using 2 μL of total RNA as input. Adapter-ligated libraries from SF were amplified with 26 PCR cycles, and libraries from serum were amplified with 23 PCR cycles, using barcoded Illumina reverse primers in combination with the Illumina forward primer (Illumina, San Diego, CA, USA). Library quality control was performed using DNA1000 Chip (Agilent, Santa Clara, CA, USA). An equimolar pool consisting of all sequencing libraries was prepared and sequenced on an Illumina Novaseq with 75 base pairs (bp) paired end reads.

#### 4.4.3. Small RNA Sequencing Data Processing and Differential Expression Analysis

Sequencing data were processed and analyzed by TAmiRNA (TAmiRNA GmbH, Vienna, Austria) following the miND pipeline, adapted to equine samples [53]. Briefly, after demultiplexing, the overall quality of the sequencing data was evaluated automatically and manually with fastQC v0.11.8 and multiQC v1.7 [54]. For miRNA analysis, reads from all quality samples were adapter trimmed, quality filtered using cutadapt v2.3 [55] and filtered for a minimum length of 17 nucleotides. Mapping steps were performed with bowtie v1.2.2 [56] and miRDeep2 v2.0.1.2 [57]. Reads were mapped first against the genomic reference EquCab.3.0 provided by Ensembl [58], allowing for two mismatches, and subsequently miRBase v22.1 [59], filtered for equine (eca) miRNAs only, allowing for one mismatch. For a general RNA composition overview, non-miRNA mapped reads were mapped against RNAcentral [60] and then assigned to various RNA species of interest. Statistical analyses of pre-processed sequencing data were performed with R v3.6 and the packages pheatmap v1.0.12 and genefilter v1.68 to generate heatmaps. PCA plots of pre-processed miRNA data were created in R v4.3 using the packages stats v3.6.2 and ggplot2 v.3.4.2. Differential expression analysis with edgeR v3.28 [61] used the quasi-likelihood negative binomial generalized log-linear model functions provided by the package. The independent filtering method of DESeq2 [62] was adapted for use with edgeR to remove low abundant miRNAs and thus optimize FDR correction.

### 4.5. Pathway Analysis and Target Prediction

Prediction of functional targets for the differentially expressed miRNAs was carried out using IPA software v22.0 (Qiagen, Manchester, UK). For this, equine miRNAs that were differentially expressed in serum and in SF were matched to their human equivalents, and the corresponding miRbase IDs were input together into IPA as identifiers. Data were analyzed using the “Target Prediction” function, and results were robustly filtered for experimentally observed targets in chondrocytes and osteoblasts. The list of predicted mRNA targets was combined with the list of differentially expressed miRNAs along with their expression values (logFC) and significance levels and input back into IPA. Data were analyzed using the “Core Analysis” function, which calculates the *p*-value of overlap between the molecules in the dataset with the disease and functions contained in the Ingenuity Knowledge Base using a right-tailed Fisher’s exact test. Results were filtered for experimentally observed associations only and used to algorithmically generate a network of canonical pathways, biological functions, diseases and network-eligible molecules based on their connectivity.

### 4.6. Selection of Differentially Expressed miRNAs for Validation

All differentially expressed miRNAs were reviewed, and a limited group was selected for validation through RT-qPCR. miRNAs were selected if they were differentially expressed (FDR < 0.05 or *p* < 0.05; Table 2) in both SF and serum with the same directionality in change, with no cut-off for logFC. Of these, four miRNAs that had been previously identified in OA-related studies were selected for validation. Two additional miRNAs (eca-miR-99b and eca-miR-146a) were also selected for further analyses. While these two molecules did not meet the previously mentioned criteria, a preliminary analysis in which data were mapped against the human genome (results not published) found hsa-miR-99b-5p and hsa-miR-146a-5p to be differentially expressed in serum and SF, and so, these molecules were selected for further validation.

### 4.7. Sample Size Calculation

To estimate the required sample size for miRNA validation, a sample size calculation was performed in R using the package pwr v1.3-0. miRNA sequencing data were input as reference for data variation and expected size effect, assuming a similar behavior of qPCR data. Significance level was set at 0.05, and statistical power was set at 95%. One-sample t-tests were performed individually for each miRNA in the selected panel for both SF and serum. The final sample size corresponded to the test that required the highest n number (i.e., if one miRNA required 20 samples for validation and the following miRNA required 30 samples, the final sample size corresponded to 30 samples).

### 4.8. RT-qPCR

Total RNA was extracted from 200 μL of SF diluted 1:1 in RNA or from 200 μL of serum using miRNeasy Serum/Plasma Advanced Kit, with the addition of RNA spike-ins (RNA Spike-in Kit, for RT, Qiagen, Manchester, UK) and glycogen. cDNA synthesis was performed using miRCURY LNA RT Kit (Qiagen, Manchester, UK) using 4 μL of total RNA as input. qPCR was performed using miRCURY^®^ LNA^®^ miRNA SYBR^®^ Green PCR (Qiagen, Manchester, UK) using 3 μL of diluted cDNA (1:30) as input on a LightCycler^®^ 96 (Roche Life Science, Penzberg, Germany). Bench-validated, equine compatible miRNA assays were obtained from Qiagen (Appendix A). Following the determination of the best normalization strategy (Appendix A), the 2^−ΔCq^ method was used for analysis of relative gene expression [63].

### 4.9. Analysis of Clinical Variables

The influence of clinical variables on miRNA expression in serum and SF was investigated through analysis of PCs using linear models or one-way ANOVAs, depending on data type. Briefly, miRNA data were normalized and scaled, and PCs were calculate using the prcomp function in R. For SF miRNA data, singular value decomposition was applied prior to PC calculation. Ordinal variables such as level of work, BCS, macroscopic score and microscopic score were assessed as numerical values. Age, level of work, BCS, macroscopic score and microscopic score were assessed against each PC using linear models; sex, group and presence of shoes were numerically encoded and assessed against each PC using one-way ANOVA tests. All *p*-values were adjusted for multiple comparisons using FDR (Benjamini–Hochberg) correction. Results were presented as a heatmap, generated with the pheatmap function in R. One-way ANOVAs followed by Tukey’s multiple comparison tests were used to compare miRNA expression between age classes.

## 5. Conclusions

OA is a heterogenous disease where a complex interplay of internal and external variables contributes to altered miRNA expression in a range of sample types. In this study, we identified a panel of miRNAs that were differentially expressed in serum and SF samples, two of which (miR-199a-3p and miR-148a) warrant further investigation. Further characterization of these molecular changes and their potential targets will contribute to a better understanding of the biological processes they regulate and could provide novel insights into the mechanisms of early OA. While clinical application is still limited, this study is one of the first steps towards the use of miRNAs as biomarkers of equine OA.

## Figures and Tables

**Figure 1 ijms-26-11190-f001:**
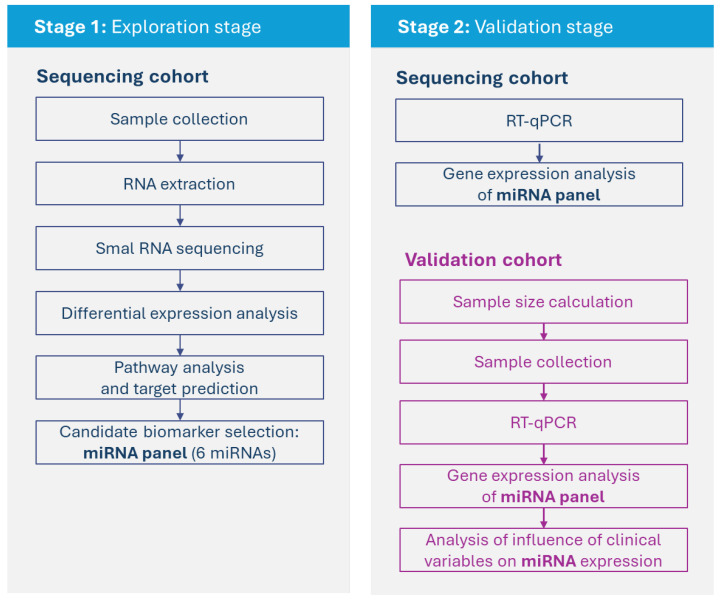
Overview of experimental set up.

**Figure 2 ijms-26-11190-f002:**
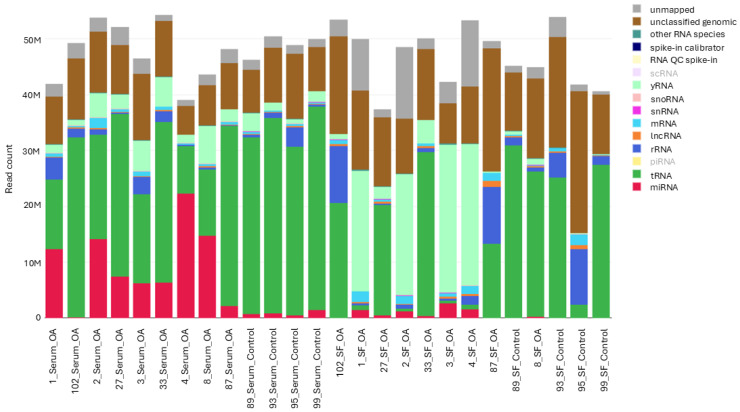
Absolute reads composition of samples in the sequencing cohort. Bar colors represent different RNA types, and labels with lower opacity correspond to RNA types that were not identified in this dataset. Each bar represents a different sample. QC, quality control; mRNA, messenger RNA; rRNA, ribosomal RNA; scRNA, small conditional RNA.

**Figure 3 ijms-26-11190-f003:**
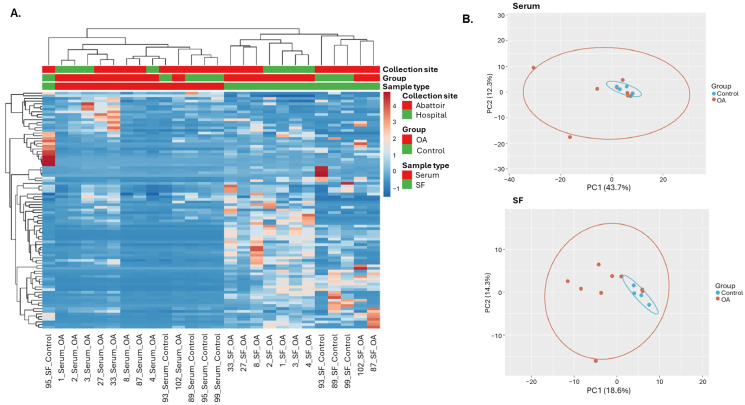
Heatmap (**A**) and PCA plots (**B**) of normalized and scaled miRNA reads. Ellipses in the PCA plots represent a confidence level of 0.95. Images were created with R v3.6 and the packages pheatmap v1.0.12, genefilter v1.68, stats v3.6.2 and ggplot2 v.3.4.2.

**Figure 4 ijms-26-11190-f004:**
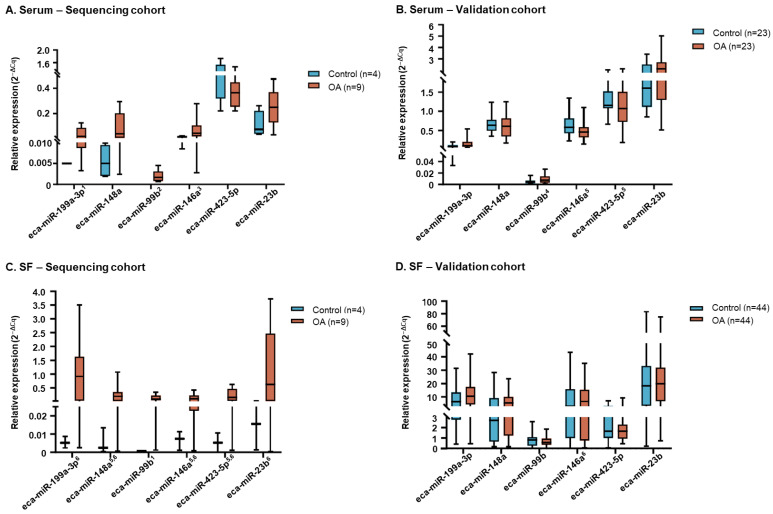
Relative expression of selected miRNAs in the (**A**) serum sequencing cohort, (**B**) serum validation cohort, (**C**) SF sequencing cohort and (**D**) SF validation cohort using RT-qPCR. ^1^ miRNA expression not detected in 3 samples in the control group. ^2^ miRNA expression not detected in any of the samples in the control group. ^3^ miRNA expression not detected in 2 samples in the control group. ^4^ miRNA expression not detected in 1 sample in the control group and 6 samples in the OA group. ^5^ miRNA expression not detected in 1 sample in the OA group. ^6^ miRNA expression not detected in 1 sample in the control group. Box plots show the interquartile data range as box boundaries, with error bars showing the minimum and maximum data range and the horizontal line showing the median. Images were created using GraphPad Prism version 8.0 for Windows. Cq, quantification cycle.

**Table 1 ijms-26-11190-t001:** Demographics and clinical characteristics of the sequencing cohort, stratified by group.

	Control (N = 4)	OA (N = 9)
**Collection site, n (%)**		
Abattoir ^1^	4 (100)	5 (55.5)
Hospital ^2^	0	4 (44.4)
**Age, years**		
n	3	9
Mean (SD)	6.3 (7.5)	6.6 (3.5)
Min; Max	2; 15	2; 14
Missing/not reported	1	0
**Sex, n (%)**		
n	3	9
Female	2 (66.7)	2 (22.2)
Male		
Neutered		6 (66.7)
Not neutered	1 (33.3)	1 (11.1)
Missing/not reported	1	0
**Breed, n (%)**		
n	3	8
Arab	1 (33.3)	0
Friesian	0	1 (12.5)
Standardbred	1 (33.3)	1 (12.5)
Swedish Warmblood	1 (33.3)	4 (50.0)
Thoroughbred	0	2 (25.0)
Missing/not reported	1	1
**Occupation, n (%)**		
n	1	3
Racing	1 (100)	3 (100)
Missing	3	6
**OA severity, n (%) ^3^**		
n	4	5
Control	4 (100)	0
Mild	0	3 (60.0)
Moderate	0	1 (20.0)
Severe	0	1 (20.0)
Missing/not reported	0	4
**Joint affected, n (%)**		
n	4	9
Carpal	4 (100)	6 (66.7)
Metacarpophalangeal	0	3 (33.3)

Percentages were calculated excluding the cases for which information was missing/not reported. ^1^ Samples collected post-mortem. ^2^ Samples collected during lameness examination. ^3^ Classification of OA severity was performed as previously described by Skiöldebrand et al. [15]. SD, standard deviation.

**Table 2 ijms-26-11190-t002:** Differentially expressed miRNAs between control and OA samples in serum and SF.

miRNA	logFC ^1^	*p*-Value	FDR	Significance
**Serum**				
eca-miR-9048	−8.74	<0.0001	0.0164	Decreased in OA
eca-miR-143	4.09	0.0001	0.0164	Increased in OA
eca-miR-25	1.99	0.0002	0.0164	Increased in OA
eca-miR-146a	2.67	0.0004	0.0242	Increased in OA
eca-miR-1291a	−7.35	0.0007	0.0242	Decreased in OA
eca-miR-8986b	−7.35	0.0007	0.0242	Decreased in OA
eca-miR-1892	−7.26	0.0008	0.0242	Decreased in OA
eca-miR-8954	−7.26	0.0008	0.0242	Decreased in OA
eca-miR-330	−6.93	0.0008	0.0242	Decreased in OA
eca-miR-490-3p	−7.47	0.0008	0.0242	Decreased in OA
eca-miR-191a	1.76	0.0010	0.0255	Increased in OA
eca-miR-345-5p	−6.91	0.0010	0.0255	Decreased in OA
eca-miR-16	2.24	0.0014	0.0296	Increased in OA
eca-miR-133a	3.80	0.0014	0.0296	Increased in OA
eca-miR-223	4.32	0.0022	0.0446	Increased in OA
eca-miR-129b-3p	−5.60	0.0033	0.0580	Decreased in OA
eca-miR-8951	−5.60	0.0033	0.0580	Decreased in OA
eca-miR-199a-3p	2.15	0.0038	0.0597	Increased in OA
eca-miR-199b-3p	2.15	0.0038	0.0597	Increased in OA
eca-miR-483	−5.01	0.0049	0.0729	Decreased in OA
eca-miR-142-5p	1.69	0.0065	0.0935	Increased in OA
eca-miR-15a	3.61	0.0076	0.1032	Increased in OA
eca-miR-148a	1.63	0.0087	0.1094	Increased in OA
eca-miR-423-5p	−1.06	0.0088	0.1094	Decreased in OA
eca-miR-23b	−1.56	0.0106	0.1271	Decreased in OA
eca-miR-93	1.77	0.0112	0.1287	Increased in OA
eca-miR-744	2.64	0.0122	0.1351	Increased in OA
eca-miR-130a	3.42	0.0132	0.1410	Increased in OA
eca-miR-8992	−3.88	0.0143	0.1444	Decreased in OA
eca-miR-8977	−5.19	0.0144	0.1444	Decreased in OA
eca-miR-423-3p	−1.12	0.0217	0.2064	Decreased in OA
eca-miR-206	3.59	0.0221	0.2064	Increased in OA
eca-miR-194	2.59	0.0227	0.2064	Increased in OA
eca-miR-1	4.53	0.0235	0.2077	Increased in OA
eca-let-7f	−1.31	0.0278	0.2358	Decreased in OA
eca-miR-30e	1.84	0.0283	0.2358	Increased in OA
eca-miR-98	−1.88	0.0292	0.2371	Decreased in OA
eca-miR-340-5p	2.30	0.0312	0.2464	Increased in OA
eca-miR-140-3p	4.07	0.0323	0.2482	Increased in OA
eca-miR-23a	−1.24	0.0340	0.2547	Decreased in OA
eca-miR-27b	1.55	0.0470	0.3437	Increased in OA
eca-miR-2483	3.24	0.0492	0.3465	Increased in OA
eca-miR-7177b	8.21	0.0497	0.3465	Increased in OA
**Synovial fluid**				
eca-miR-324-5p	−6.66	<0.0001	<0.0001	Decreased in OA
eca-miR-296	−5.98	<0.0001	0.0002	Decreased in OA
eca-miR-615-5p	−9.25	0.0001	0.0072	Decreased in OA
eca-miR-671-3p	−4.69	0.0004	0.0187	Decreased in OA
eca-miR-27a	−4.42	0.0005	0.0187	Decreased in OA
eca-miR-184	−4.47	0.0006	0.0187	Decreased in OA
eca-miR-1291a	−9.55	0.0006	0.0187	Decreased in OA
eca-miR-148a	2.45	0.0026	0.0646	Increased in OA
eca-miR-423-5p	−1.90	0.0032	0.0646	Decreased in OA
eca-miR-23b	−3.04	0.0032	0.0646	Decreased in OA
eca-miR-598	−7.76	0.0059	0.1090	Decreased in OA
eca-miR-206	−7.24	0.0075	0.1270	Decreased in OA
eca-miR-199a-3p	1.63	0.0122	0.1770	Increased in OA
eca-miR-199b-3p	1.63	0.0122	0.1770	Increased in OA
eca-miR-31	−6.76	0.0149	0.1950	Decreased in OA
eca-miR-92b	−2.66	0.0153	0.1950	Decreased in OA
eca-miR-99a	4.81	0.0169	0.2020	Increased in OA
eca-miR-1892	−6.60	0.0360	0.3930	Decreased in OA
eca-miR-10b	0.89	0.0368	0.3930	Increased in OA
eca-miR-27b	−2.15	0.0437	0.4350	Decreased in OA
eca-miR-151-5p	10.72	0.0465	0.4350	Increased in OA
eca-miR-342-5p	10.82	0.0480	0.4350	Increased in OA
eca-miR-211	10.67	0.0495	0.4350	Increased in OA

^1^ A negative logFC corresponds to decreased miRNA expression in the OA group compared with the control group. logFC, log fold change.

**Table 3 ijms-26-11190-t003:** Demographics and clinical characteristics of the validation cohort, stratified by sample type and group.

	Serum		SF	
	Control (N = 23)	OA (N = 23)	Control (N = 44)	OA (N = 44)
**Collection site, n (%)**				
Abattoir ^1^	0	0	39 (88.6)	40 (90.9)
Hospital/Clinic ^2^	23 (100)	23 (100)	5 (11.4)	4 (9.1)
A	23 (100)	9 (39.1)	0	0
B	0	5 (21.7)	5 (100)	4 (100)
C	0	9 (39.1)	0	0
**Age, years**				
n	23	9	41	41
Mean (SD)	9.8 (4.2)	11.1 (5.4)	10.2 (6.8)	15.7 (6.7)
Min; Max	3; 18	4; 23	2; 20	3; 25
Missing/not reported	0	14	3	3
*p*-value	0.6582 ^1^		0.0003 ^1^	
**Sex, n (%)**				
n	23	14	25	26
Female	8 (34.8)	2 (14.3)	17 (68.0)	8 (30.8)
Neutered male	15 (65.2)	12 (85.7)	8 (32.0)	181 (69.2)
*p*-value	0.2603 ^2^		0.0227 ^2^	
Missing/not reported	0	9	19	18
**Body Condition Score, n (%)**				
n	23	9	0	0
1–3	0	0	–	–
4	0	1 (11.1)	–	–
5	19 (82.6)	6 (66.6)	–	–
6	3 (13.0)	2 (22.2)	–	–
7	1 (4.3)	0	–	–
8–9	0	0	–	–
Missing/not reported	0	14	44	44
**Breed, n (%)**				
n	23 ^3^	23 ^3^	19 ^4^	16 ^4^
Appaloosa	0	1 (4.3)	0	0
Cob ^5^	1 (4.3)	0	2 (10.5)	1 (6.3)
Connemara ^5^	4 (17.4)	1 (4.3)	0	0
Dales ^5^	0	1 (4.3)	0	0
Dutch Warmblood	0	1 (4.3)	0	0
Hanoverian	0	2 (8.7)	0	0
Holsteiner	0	1 (4.3)	0	0
Irish cob	0	1 (4.3)	0	0
Irish Draught	2 (8.7)	0	0	0
Irish Sport Horse ^5^	9 (39.1)	4 (17.4)	2 (10.5)	3 (18.8)
Lusitano	1 (4.3)	0	0	1 (6.3)
Pony	1 (4.3)	1 (4.3)	4 (21.4)	0
Thoroughbred ^5^	2 (8.7)	9 (39.1)	7 (36.8)	9 (56.3)
Warmblood ^5^	1 (4.3)	0	0	2 (12.5)
Welsh Pony/Cob ^5^	2 (8.7)	1 (4.3)	4 (21.4)	0
Missing/not reported	0	0	25	25
**Occupation, n (%) ^3^**				
n	23	18	0	0
Not in work (‘out in the field’)	2 (8.7)	2 (11.1)	–	–
All-rounder	5 (21.7)	1 (5.6)	–	–
Dressage	1 (4.3)	0	–	–
Eventing	2 (8.7)	0	–	–
Hacking	5 (21.7)	32 (16.5)	–	–
Hunting	3 (13.0)	1 (5.6)	–	–
Leisure	1 (4.3)	0	–	–
Racing	0	9 (50.0)	–	–
Schooling	4 (17.4)	1 (5.6)	–	–
Showjumping	0	1 (5.6)	–	–
Missing/not reported	0	5	44	44
**Current level of work** **(0–3), n (%) ^5^**				
n	23	18	0	0
0 (‘out in the field’)	2 (8.7)	2 (11.1)	–	–
1 (‘light work’)	12 (52.2)	4 (22.2)	–	–
2 (‘medium work’)	7 (30.4)	2 (11.1)	–	–
3 (‘intense work’)	2 (8.7)	10 (55.6)	–	–
Missing/not reported	0	10	44	44
*p*-value	0.0214 ^6^		–	
**Shoes, n (%)**				
n	18	8	0	0
All four feet	13 (72.2)	3 (37.5)	n/a	n/a
Front feet	2 (11.1)	1 (12.5)	n/a	n/a
Unshod	3 (16.7)	4 (50.0)	n/a	n/a
Missing/not reported	5	15	44	44
**Joints affected, n (%) ^7,8^**				
Distal interphalangeal	–	2 (9.1) ^3^	0	0
Intervertebral	–	2 (9.1)	0	0
Metacarpophalangeal	–	7 (31.8) ^3^	44 (100) ^8^	44 (100) ^8^
Metatarsophalangeal	–	4 (18.1) ^3^	0	0
Sacroiliac	–	2 (9.1) ^3^	0	0
Scapulohumeral	–	1 (4.5) ^3^	0	0
Tarsometatarsal	–	2 (9.1) ^3^	0	0
Front limb ^9^	–	2 (9.1) ^3^	0	0
**Joint gross/macroscopic score, n (%) ^10^**				
n	0	0	44	44
0	–	–	21 (47.7)	0
1	–	–	23 (52.3)	0
2	–	–	0	10 (22.7)
3	–	–	0	16 (36.4)
4	–	–	0	11 (25.0)
5	–	–	0	4 (9.1)
6	–	–	0	2 (4.5)
7	–	–	0	1 (2.3)
8–9	–	–	0	0
Mean (SD)	–	–	0.5 (0.5)	3.4 (1.2)
Min; Max	–	–	0; 1	2; 7
*p*-value	–		<0.0001 ^6^	
Missing/not reported	23	23	0	0
**Articular cartilage** **microscopic score, n (%) ^11^**				
n	0	0	34	30
0	–	–	0	0
1	–	–	5 (14.7)	0
2	–	–	8 (23.5)	5 (16.7)
3	–	–	7 (20.6)	4 (13.3)
4	–	–	9 (26.5)	5 (16.7)
5	–	–	2 (5.9)	5 (16.7)
6	–	–	1 (2.9)	5 (16.7)
7	–	–	1 (2.9)	4 (13.3)
8	–	–	1 (2.9)	1 (3.3)
9–15	–	–	0	0
16	–	–	0	1 (3.3)
17–20	–	–	0	0
Mean (SD)	–	–	3.2 (1.7)	5.0 (2.9)
Min; Max	–	–	1; 8	2; 16
*p*-value	–		0.0015 ^6^	
Missing/not reported	23	23	10	14

Percentages were calculated excluding the cases for which information was missing/not reported. ^1^ Calculated using Mann–Whitney U test. ^2^ Calculated suing Fisher’s exact test. ^3^ As reported by owner. ^4^ As reported in the passport. ^5^ Including breed crosses. ^6^ Calculated using chi-square test for trend. ^7^ Horses could be included in more than one category. ^8^ For SF, samples were only collected from the metacarpophalangeal joint; information regarding other joints was not available. ^9^ Owner could not recall affected joint but reported the affected limb. ^10^ Assessed at post-mortem. ^11^ Histologic/microscopic scoring was carried out using OARSI microscopic grading system [16]. –, not applicable/available; BCS, body condition score. Subheaders are in bold.

## Data Availability

Data has been submitted to NCBI’s gene expression omnibus (GEO).

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
