# Peer review of "Synovial Fluid and Serum MicroRNA Signatures in Equine Osteoarthritis"

_ijms, 2025, doi:10.3390/ijms262211190_

Round 1
Reviewer 1 Report
Comments and Suggestions for Authors
This manuscript describes microRNA signatures detected in equine serum and synovial fluid samples and compares differential miRNA expression between healthy and osteoarthritis samples. A strength of the study is the 2-stage design in which small RNAsequencing is employed with a relatively small cohort of samples (n=13) in an exploratory stage, followed by validation in both the sequencing cohort in addition to a larger RT-qPCR validation cohort (n=88 synovial fluid and n=46 serum). The study provides valuable descriptive data about the abundances of small RNAs in equine serum and synovial fluid and is a good initial step in attempting to identify miRNAs that could serve as potential biomarkers for the early diagnosis of OA in horses.
While the sampling limitations, especially for obtaining synovial fluid (SF) from healthy animals are inherent to any study, the authors do a fairly good job of addressing potential confounders such as whether the sample was collected ante- or postmortem in their analyses, while also analyzing other covariates, such as exercise intensity, in the larger validation cohort.
While addressed well in the data analysis section, there are a few discrepancies in the demographic data that need to be clarified. In addition, the authors should expand upon the limitations of using abattoir-derived samples in that gross assessments can be performed, but clinical data such as lameness cannot be evaluated. The authors briefly mention that differences between live and abattoir-derived samples may have influenced the results, but there seems to be an opportunity for the authors to specifically comment on this as a variable in their study since both types of samples were included in the OA group for the sequencing cohort and for both Control & OA groups in the validation cohort.
The figures and tables are generally well prepared, but there are some opportunities for additional clarification or improvement:
Figure 2. Could the authors label or identify the individual samples as healthy or OA? There appears to be clustering of some sample compositions, but the current sample labels are not very informative. Can the authors also comment on the substantial variation in absolute reads composition—is this much variation in composition anticipated between SF and serum samples? Are there any variables that might help account for what appears to be some clustering within both serum and SF samples? I would also be curious if any of the clustering was a result of ante- vs. postmortem collection.
Figure 3. There appears to be fairly discrete clustering by biofluid between serum and SF samples, with the exception of “959F @ Synovial fluid”. Was there anything unique about this sample? Was this an abattoir-derived sample since it was in the PM Cohort?
Table 1 indicates that 3 control horses (100%) were racehorses, but the study reports 4 Controls. In addition, of the 3 racehorses, breeds were described as Arabian & Missing with only 1 typical racing breed (Standardbred). If breed was ‘Missing’, how could this horse be identified as a racehorse? Unless I’m misunderstanding something, it appears that 1 horse is still unaccounted for in the Control group for Age, Sex, Breed & Occupation (but not OA severity or Collection site where n=4).
Table 3. Is there any data available on prior treatment or medication in the OA group undergoing lameness exam that could have influenced miRNA expression?
Page 5, Lines 136-137: The authors should expand upon their description of what SVD with imputation was used and how outliers were handled.
Figure S1: Is labeled serum sample but includes both serum & synovial fluid samples.
Supplemental Figures: Many of the axes labels for the supplemental figures are quite small & difficult to read.
Author Response
|
Point-by-point response to Comments and Suggestions for Authors |
|
General comment: This manuscript describes microRNA signatures detected in equine serum and synovial fluid samples and compares differential miRNA expression between healthy and osteoarthritis samples. A strength of the study is the 2-stage design in which small RNAsequencing is employed with a relatively small cohort of samples (n=13) in an exploratory stage, followed by validation in both the sequencing cohort in addition to a larger RT-qPCR validation cohort (n=88 synovial fluid and n=46 serum). The study provides valuable descriptive data about the abundances of small RNAs in equine serum and synovial fluid and is a good initial step in attempting to identify miRNAs that could serve as potential biomarkers for the early diagnosis of OA in horses. While the sampling limitations, especially for obtaining synovial fluid (SF) from healthy animals are inherent to any study, the authors do a fairly good job of addressing potential confounders such as whether the sample was collected ante- or postmortem in their analyses, while also analyzing other covariates, such as exercise intensity, in the larger validation cohort. While addressed well in the data analysis section, there are a few discrepancies in the demographic data that need to be clarified. In addition, the authors should expand upon the limitations of using abattoir-derived samples in that gross assessments can be performed, but clinical data such as lameness cannot be evaluated. |
|
Comment 1: The authors briefly mention that differences between live and abattoir-derived samples may have influenced the results, but there seems to be an opportunity for the authors to specifically comment on this as a variable in their study since both types of samples were included in the OA group for the sequencing cohort and for both Control & OA groups in the validation cohort. |
|
Response 1: We thank the Reviewer for this comment. In line with this suggestion, we have added the following information to the Discussion: Lines 533–543: “There were also limitations associated with the sample collection and group allocation. For abattoir-derived samples, macroscopic and microscopic joint assessments were performed, yet clinical data such as lameness could not be evaluated. The relationship between joint macroscopic changes and overt clinical signs remains uncertain, and in abattoir specimens where clinical history is lacking, we cannot reliably assess how those macroscopic changes align with clinical lameness. Additionally, there is a potential for degradation and/or altered biochemical profiles in abattoir-derived samples, which can also affect the results. On the other hand, it is not possible to collect information regarding macroscopic and microscopic joint assessments for excess clinical samples. All of these issues can lead to inconsistencies in how the samples are al-located to the different groups and potentially lead to different subpopulations within the control and OA groups.” |
|
Comment 2: The figures and tables are generally well prepared, but there are some opportunities for additional clarification or improvement: Figure 2: A) Could the authors label or identify the individual samples as healthy or OA? There appears to be clustering of some sample compositions, but the current sample labels are not very informative. B) Can the authors also comment on the substantial variation in absolute reads composition—is this much variation in composition anticipated between SF and serum samples? C) Are there any variables that might help account for what appears to be some clustering within both serum and SF samples? I would also be curious if any of the clustering was a result of ante- vs. postmortem collection. |
|
Response 2: A) The labels in Figure 2 have been edited, and the group classification has been added to each specific sample, for clarity.
B) Our group has previously reported a wide variety of small RNA molecules in equine serum and SF samples [1,2], and so the variation in absolute read composition observed in the present study was anticipated. As mentioned in the Discussion,we hypothesize that the differences in relative abundance of RNA species among sample types might be related to the fact that the molecular content in serum is derived from numerous tissues, while the SF content mainly originates from joint tissues, leading to different molecular profiles. In line with this comment and with comment 5 from Reviewer 2, we expanded the information we had previously included, as follows: Lines 390–399: “SF is the biofluid which more closely reflects the local processes in the joint, whereas serum provides an overview of systemic events.Circulating miRNAs in blood are attractive candidates as biomarkers of disease because samples can generally be obtained through minimally invasive techniques and provide an overview of systemic events. On the other hand, SF is a plasma ultrafiltrate that closely reflects the local processes in the joint due to its biological and functional proximity with the different joint tissues [18]. This, along with the fact that SF can be sampled with relative ease from most equine synovial joints, renders SF analysis a valuable diagnostic tool for OA in clinical settings [16]. To compare the molecular patterns between SF and serum, matched samples were collected for each horse in the sequencing cohort.”
C) The present study focused on the analysis of the miRNA profile, and therefore clustering analysis was not performed for the overall reads but rather for miRNAs reads specifically, as reported in Figure 3. The top dendrogram of Figure 3A (which illustrates the hierarchy of samples with similar miRNA expression) suggests that samples generally clustered by sample type (serum vs SF), but not by group (OA vs control) or collection site (abattoir vs hospital). Note that we have edited the labels in Figure 3A for clarity.
References: 1. Castanheira, C., et al. BMC Vet Res 2021, 17, 26. 2. Baker, M.E., et al. Int J Mol Sci 2022, 23. 16. McIlwraith, C.W. et al. Osteoarthritis Cartilage 2010, 18, S93–S105. 18. Steel, C. M. Vet Clin North Am Equine Pract 2008, 24, 437–454. |
|
Comment 3: Figure 3. There appears to be fairly discrete clustering by biofluid between serum and SF samples, with the exception of “959F @ Synovial fluid”. Was there anything unique about this sample? Was this an abattoir-derived sample since it was in the PM Cohort? |
|
Response 3: As mentioned in the answer to Comment 2, Figure 3A showed that samples generally clustered by sample type (serum vs SF), with the exception of sample 95SF. We hypothesize that this was due to the percentage of unclassified reads in this sample which was the highest of all samples analysed (61% in sample 95SF vs 17–44% in other SF samples). If a substantial portion of reads are unclassified, data composition and potentially sample clustering will be affected. While this was an abattoir-derived sample, Figure 3A suggests that sample site collection alone cannot justify the sample clustering, as evidenced by the fact that other abattoir-derived SF samples clustered with hospital-derived samples. It is possible that technical factors such as sample degradation or pre-analytical sample processing (leading to variable cellular content) might have played a role. The information available at the time was insufficient to help us determine the reason behind these changes. In line with this feedback, we have added some information to the text to further contextualize these differences. Lines 149–154: “A heatmap was built on reads per million (RPM) normalized and scaled reads using the unit variance method for visualization in heatmaps (Figure 3A). This unsupervised analysis demonstrated a generally distinct miRNA expression pattern between serum and SF samples, with all but one sample (95SF) clustering according to sample type. Of note, 95SF was the sample with the highest percentage of unclassified genome (Figure 2), which could have affected data sample clustering.” |
|
Comments 4: Table 1 indicates that 3 control horses (100%) were racehorses, but the study reports 4 Controls. In addition, of the 3 racehorses, breeds were described as Arabian & Missing with only 1 typical racing breed (Standardbred). If breed was ‘Missing’, how could this horse be identified as a racehorse? Unless I’m misunderstanding something, it appears that 1 horse is still unaccounted for in the Control group for Age, Sex, Breed & Occupation (but not OA severity or Collection site where n=4). |
|
Response 4: Thank you for highlighting this issue. We have reviewed and updated the demographic information for the horses included in the sequencing cohort, specifically regarding breed, occupation and sex. |
|
Comment 5: Table 3. Is there any data available on prior treatment or medication in the OA group undergoing lameness exam that could have influenced miRNA expression? |
|
Response 5: Unfortunately, this information was not available for the animals included in this study. However, we agree this would be valuable information, and that it should be taken into consideration in future studies. In line with this comment, we have added the following text to the Discussion: Lines 546–548: “Furthermore, there was no information available on prior treatment/medication, and its potential effect on miRNA expression could not be assessed.” |
|
Comment 6: Page 5, Lines 136-137: The authors should expand upon their description of what SVD with imputation was used and how outliers were handled. |
|
Response 6: For the PCA plots in Figure 3, PCs were calculated based on miRNA that had been previously normalized and scaled. For this reason, PCs were calculated using the function prcomp() in R (part of the base stats package, which uses singular value decomposition), and visualized using the package ggplot2 v.3.4.2. All samples were retained to reflect the full data structure, and no specific outlier removal processes were applied. We have edited the text to reflect this information, as follows: Lines 155–161: “Principal component analyses (PCA) was were performed using RPM normalized miRNA reads, which were scaled using unit variance. Principal components (PCs) were calculated using singular value decomposition with imputation (Figure 3B)using the function prcomp() in R (package stats v3.6.2) and visualized using the package ggplot2 v.3.4.2 (Figure 3B). All samples were retained to reflect the full data structure, and no specific outlier removal processes were applied. The PCA plots showed an overlap between the OA and control groups for both serum and SF.” |
|
Comment 7: Figure S1: Is labeled serum sample but includes both serum & synovial fluid samples. |
|
Response 7: Thank you for highlighting this issue. We have now edited the title of the figure to: “Figure S1. Relative reads composition of all serum and synovial fluid samples.” |
|
Comment 8: Supplemental Figures: Many of the axes labels for the supplemental figures are quite small & difficult to read. |
|
Response 8: In line with your feedback, we have edited the labels of the supplementary figures and increased the font size for clarity. |

Reviewer 2 Report
Comments and Suggestions for Authors
The manuscript discusses an intriguing and significant topic: understanding the pathophysiological mechanisms of OA in horses and the role of microRNA in these processes. While it presents many results, there are some shortcomings. In general, the manuscript is difficult to follow and remains unclear to the reader after the first reading. This may be due to the extensive detail in the results and the structure of the materials and methods section. I recommend that the authors present parts of the materials and methods visually to enhance clarity and minimize repetition. Additionally, organizing the materials and methods section chronologically—beginning with a description of the animals, the study's inclusion criteria, and the group formation criteria—would improve coherence.
The results are overly detailed and could be summarized more clearly for better understanding.
I recommend rewriting the introduction in Lines 82-88 to emphasize the study's objectives instead of the results. Additionally, the capitalization of 'Control group' throughout the text seems unnecessary. In the Discussion section, particularly in Lines 339-340, it would be beneficial to emphasize the significance of comparing molecules in serum and synovial fluid, noting that synovial fluid is an ultrafiltrate of plasma.
I believe that after revisions, the manuscript will meet all the criteria for publication.
Author Response
|
Comment 1: The manuscript discusses an intriguing and significant topic: understanding the pathophysiological mechanisms of OA in horses and the role of microRNA in these processes. While it presents many results, there are some shortcomings. In general, the manuscript is difficult to follow and remains unclear to the reader after the first reading. This may be due to the extensive detail in the results and the structure of the materials and methods section. I recommend that the authors present parts of the materials and methods visually to enhance clarity and minimize repetition. Additionally, organizing the materials and methods section chronologically—beginning with a description of the animals, the study's inclusion criteria, and the group formation criteria—would improve coherence. The results are overly detailed and could be summarized more clearly for better understanding. |
|
Response 1: We thank you for your comment. As suggested, we have revised the manuscript and incorporated some changes to the text, specifically: we have simplified and summarised the Results section, which included moving some content to Appendix A; we restructured the Methods section to improve clarity and summarized the information, with some content moved to Appendix B; we edited Figure 1 from an overview of methodology to an overview of the experimental set up, to provide additional detail on the steps followed; and we edited some parts of the Discussion to improve clarity. All changes were tracked and can be found throughout the manuscript. |
|
Comment 3: I recommend rewriting the introduction in Lines 82-88 to emphasize the study's objectives instead of the results. |
|
Response 3: The final paragraph of the Introduction has been edited as follows: Lines 81–89: “The objective of the present study was to determine the miRNA profile of serum and SF of horses affected by OA and control horses, identify differentially expressed miRNAs between the two groups in equine serum and SF which have and determine their potential as biomarkers of OA.For this, we used a small RNA sequencing approach to investigate the miRNA profile of matched (i.e., from the same horse) equine serum and SF in OA. Additionally, we used bioinformatic tools to explore associated pathways and potential mRNA targets. Finally, we validated our findings in a larger, independent cohort and assessed the influence of different clinical variables on the expression of a selected group of miRNAs.” |
|
Comment 4: Additionally, the capitalization of 'Control group' throughout the text seems unnecessary. |
|
Response 4: Thank you for this suggestion; we have edited the text accordingly. |
|
Comment 5: In the Discussion section, particularly in Lines 339-340, it would be beneficial to emphasize the significance of comparing molecules in serum and synovial fluid, noting that synovial fluid is an ultrafiltrate of plasma. |
|
Response 5: In line with this comment, we have added the following information to the Discussion: Lines 390–399: “SF is the biofluid which more closely reflects the local processes in the joint, whereas serum provides an overview of systemic events.Circulating miRNAs in blood are attractive candidates as biomarkers of disease because samples can generally be obtained through minimally invasive techniques and provide an overview of systemic events. On the other hand, SF is a plasma ultrafiltrate that closely reflects the local processes in the joint due to its biological and functional proximity with the different joint tissues [18]. This, along with the fact that SF can be sampled with relative ease from most equine synovial joints, renders SF analysis a valuable diagnostic tool for OA in clinical settings [16]. To compare the molecular patterns between SF and serum, matched samples were collected for each horse in the sequencing cohort.” |

Round 2
Reviewer 2 Report
Comments and Suggestions for Authors
I would like to thank the authors for their efforts and adopted comments. I believe that the work is suitable for publication.